# Identification and Validation of a Chromosome 4D Quantitative Trait Locus Hotspot Conferring Heat Tolerance in Common Wheat (*Triticum aestivum* L.)

**DOI:** 10.3390/plants11060729

**Published:** 2022-03-09

**Authors:** Lu Lu, Hui Liu, Yu Wu, Guijun Yan

**Affiliations:** 1UWA School of Agriculture and Environment, The University of Western Australia, 35 Stirling Highway, Perth, WA 6009, Australia; lu.lu@research.uwa.edu.au; 2The UWA Institute of Agriculture, The University of Western Australia, 35 Stirling Highway, Perth, WA 6009, Australia; 3Chengdu Institute of Biology, Chinese Academy of Sciences, Chengdu 610041, China; wuyu@cib.ac.cn

**Keywords:** heat tolerance (HT), seedling stage, recombinant inbreed lines (RILs), single-nucleotide-polymorphism (SNP) markers, quantitative trait locus/loci (QTL)

## Abstract

Understanding of the genetic mechanism of heat tolerance (HT) can accelerate and improve wheat breeding in dealing with a warming climate. This study identified and validated quantitative trait loci (QTL) responsible for HT in common wheat. The International Triticeae Mapping Initiative (ITMI) population, recombinant inbreed lines (RILs) derived from a cross between Synthetic W7984 and Opata M85, was phenotyped for shoot length, root length, whole plant length under heat stress and corresponding damage indices (DIs) to compare HT performances of individuals. Wide variations among the RILs were shown for all the traits. A total of 13 QTL including 9 major QTL and 4 minor QTL were identified, distributed on 6 wheat chromosomes. The six major QTL with the highest R^2^ were associated with different traits under heat stress. They were all from Opata M85 background and located within a 2.2 cm interval on chromosome 4D, making up a QTL hotspot conferring HT in common wheat. The QTL hotspot was validated by genotyping-phenotyping association analysis using single-nucleotide-polymorphism (SNP) assays. The QTL, especially the 4D QTL hotspot identified and validated in this study, are valuable for the further fine mapping and identification of key genes and exploring genetic mechanism of HT in wheat.

## 1. Introduction

Heat has become a serious constraint for wheat production with global climate change [1]. Its damage may happen in arid, semiarid, tropical, and subtropical regions of the world [2]. Each 1 °C rise in temperature above the optimum can cause a 3–5% reduction in grain weight under both controlled environments [2] and field conditions [3]. Moreover, heat stress may affect not only harvest and yield but also grain quality [4]. Though reproductive stage is most sensitive for final yield and the most studied stage [5], heat damage may happen during different growth and developmental stages in the crop including seedling stage [6].

Heat tolerance (HT) in plants is a quantitative trait involving complex genetic, physiological, and biochemical controls and is affected by environmental factors. In response to heat stress, tolerant varieties generally activate an antioxidant defense system, express heat shock proteins (HSPs) and reduce senescence by staying green [7]. Genotype evaluation at different stages is necessary to better understand the HT mechanism and to breed tolerant cultivars. Grain yield is significantly correlated with seedling traits such as root length and shoot length [8]. If the plant does not survive at a seedling stage, there will be no harvest. Seedlings have greater stress avoidance and resilience than reproductive organs [9], and seedling establishment often has large impact on crop yield [10]. Some genes that contribute to seedling HT may also contribute to later stage tolerance [11]. In response to heat stress, heat shock proteins play important roles at both seedling and reproductive stages [12]. Moreover, some reports indicated that some particular QTLs at seedling stage and reproductive stage were collocated on the same chromosomes [13], and some well-known heat-induced genes at reproductive stage were found significantly enhanced in acetylation levels in wheat seedlings [14]. Therefore, QTL identification and validation for seedling stage may help us to understand the mechanism of heat tolerance at different stages.

Early-stage screening for seedlings began to attract the attention of scientists and breeders to study wheat tolerance under heat stress. However, in the limited number of such studies, the target traits were mainly focused on particular physiological indices such as chlorophyll content and chlorophyll fluorescence parameters [15] or on traits of seedling shoot or leaf [6]. Root traits have seldom been studied for heat tolerance in wheat. Vigor and deep root systems are believed to contribute to water-deficit resistance [16]. Long root and shoot lengths at the seedling stage were reported to be highly correlated with high grain yield in lentil [17]. Initial root parameters and above-ground biomass were also reported to be positively correlated in wheat [16]. Genotypes with deep roots are able to extract water from lower soil by making use of water lost in the form of deep percolation [11].

Moreover, it was observed in our previous study on wheat cultivars that the seedling lengths reduced considerably under heat stress compared to the controls [18]. Meanwhile, the root length was found to be decreased the most with variation among varieties under heat stress compared to other root traits such as root surface area, root diameter, and root volume, etc., on an overall basis. Therefore, length-related traits were selected for measurement and comparison.

The genetic mechanism of seedling tolerance to heat stress remains largely unknown so far, and the reported study of HT in wheat was mainly focused on reproductive and grain-filling stages [19]. To date, QTL mapping for wheat response to heat has identified several QTL for yield and key morpho-physiological characteristics, for example, Yang et al. [20] found QTL on the short arms of chromosomes 1B and 5A linked to grain-filling duration. Mason et al. [21,22] reported several QTL for heat susceptibility indices and yield traits on chromosomes 1A, 1B, 2A, 2B, 3B, 5A, and 6D. Paliwal et al. [23] reported QTL on chromosome 2B, 7B, and 7D for thousand grain weight, grain fill duration, and canopy temperature depression, respectively. Vijayalakshmi et al. [24] reported QTL on chromosomes 2A, 3A, 4A, 6A, 6B, and 7A with significant effects on grain yield, grain weight, grain filling, stay green, and senescence-associated traits under post-anthesis high temperature stress in wheat. All these studies focused on the reproductive stage, and there has rarely been reports for HT at seedling stage. Crop improvement demands an extensive search for genetic variability and comprehensive understanding of genetic mechanism, so traits conferring HT must be explored thoroughly to maximize germplasm exploitation [25]. The identification of QTL or genes related to HT during wheat seedling stage may help dissect the molecular mechanism of HT in wheat through a comparative study of the plant response to heat at seedling and reproductive stages. International Triticeae Mapping Initiative (ITMI) mapping population derived from synthetic hexaploid wheat ‘W7984′ and bread wheat cultivar ‘Opata’ is available at the GrainGenes database, and it is a useful resource to identify QTL [26]. This population facilitated the development of the first RFLP [27] and SSR [28] linkage maps in wheat. It has been used for QTL identification of different abiotic stresses such as drought and waterlogging, etc., in wheat [29,30,31].

The objectives of this study are the following: (1) to identify the QTL for HT at seedling stage in common wheat through genotype-phenotype association analyses; (2) to validate the identified major QTL with other genotyping markers such as SNPs; and (3) to pave the way for further fine mapping and understanding of the HT mechanism in wheat.

## 2. Results

### 2.1. Seedling Length Reduced under Heat Stress

Seedling length including root, shoot, and whole plant length were all adversely impacted by heat stress. A plant with comparatively longer length of the shoot and roots under stress was considered more heat-tolerant. The minimum, maximum, and average of length measurements in the population were all reduced under heat stress compared to the control (Figure 1). The shoot, root, and whole seedling length ranged from 8.7 cm to 20.73 cm, 5.93 cm to 16.73 cm, and 15 cm to 35.83 cm, respectively, under control conditions. Their corresponding length ranges changed to 0.7–18.7 cm, 0.73–14.67 cm, and 1.43–32.87 cm, respectively, under heat stress conditions. The average length of shoot, root, and whole seedling lengths reduced from 15.13 cm, 13.21 cm, and 28.32 cm under control conditions to to 9.34 cm, 7.78 cm, and 17.19 cm under heat stress. Larger ranges of shoots, roots, and whole seedling lengths under stress than under control suggested greater variations in the population under heat stress.

### 2.2. High Heritability of Seedling Length Traits under Heat Stress

Genetic variance for heat-tolerance-related traits in the population were significant (*p* < 0.01) regardless if they were under control or under heat-stress condition (Table 1). The heritability of shoot length, root length, and whole length under control conditions were 0.42, 0.52, and 0.51. Compared to the heritability under normal conditions, the heritability of shoot length, root length, and whole length under heat stress were much higher, which were 0.72, 0.74, and 0.78, respectively, all above 0.7.

### 2.3. Phenotypic Variation in the Traits Were All Normally Distributed

The longer length and smaller Dis (Damaging indices, its calculation and meaning were explained in Section 4.3) indicated the more tolerance against heat in parent Opata M85 than in parent Synthetic W7984. Wide variations were shown among the RILs not only for the length traits but also their derived Dis. Under stress, the shoot, root, and whole length ranged from 0.7 cm to 18.7 cm, 0.73 cm to 14.67 cm, and 1.43 cm to 32.87 cm, respectively. DIs of shoot, root, and whole length ranged from −0.31 to 0.95, −0.09 to 0.94, and −0.19 to 0.94, respectively. The phenotypic distributions (Figure 2) indicated transgressive segregations in both directions outside the ranges of the parents, suggesting polygenic inheritance nature of these traits in common wheat [32]. The two parents had the most differences on root-related traits, namely, the root length damage index (RLI) and the root length (RL). It was shown from the histogram chart that the phenotypic variation in three length traits and their Dis were all approximately normally distributed, so they were suitable for QTL analysis.

### 2.4. QTL Analysis Revealed Thirteen QTL and Six of Them Located on a Hotspot of Chromosome 4D Associated with HT

By genotype-phenotype association analysis, QTL related with HT at seedling stage were identified (Table 2). In total, 13 QTL were identified related to seedling HT (for first 6 traits), including 9 major QTL (R^2^ > 10%) and 4 minor QTL (R^2^ < 10%). They were located on chromosomes 5A, 6B, 2D, 4D, 3D, and 6D, respectively. Ten of them were contributed by the tolerant parent Opata M85, and three were contributed by the susceptible parent Synthetic W7984. Six QTL (marked in bold in Table 2) located on chromosome 4D were prominent with the highest R^2^ (from 0.26 to 0.33), which means they explained higher phenotypic variation compared to the other major and minor QTL. At the same time, they were located in close proximity of chromosome 4D (positioned from 16 to 18.2 cm) and involved in all the traits related to heat stress. Under heat stress conditions, six major 4D QTL were identified for seedling lengths and their damage indices. In contrast, under the control condition, no QTL on chromosome 4D was identified for shoot length, root length, nor whole plant length. This outcome shows that the QTL cluster found on chromosome 4D is uniquely associated with heat stress. One interesting result is that, under control conditions, all the identified QTL associated with plant lengths were donated by the maternal parent Synthetic, while under heat stress conditions, the QTL were donated by both parents, and the major QTL with the largest LOD score and R^2^ was contributed by the male parent Opata.

### 2.5. Validation for the QTL Hotspot

The QTL hotspot consisting of six QTL was focused on for validation. Of the 15,314 SNPs on the array for the RIL population, 24 SNPs were on chromosome 4D, 11 out of the 24 SNPs were within the identified QTL hotspot and distinguished 2 parents by polymorphic alleles, and 4 SNPs out of 11 could further distinguish progeny lines into 2 groups (Appendix A). The four markers are marker1 “scaffold72468_211254”, marker2 “scaffold72468_211948”, marker3 “scaffold4109_249224”, and marker4 “scaffold38811_2394108” (For details of the validation protocol, please refer to 4.5). Progeny lines with the same allele (e.g., A for marker 1 in Figure 3) as tolerant parent Opata 85 were classified into a positive group. Progeny lines with the same allele (e.g., G for marker 1 in Figure 3) as susceptible parent Synthetic W7984 were classified into a negative group. Specifically, 44 lines in the population were classified into the positive group (with allele A or C) and 54 lines were classified into the negative group (with allele G or A) by both marker 1 and marker 2. The positive group was further validated by phenotype with significant (*p* ≤ 0.01) longer length and smaller DIs (Damage indices) than the negative group. The differences were significant for all the traits, and were reflected on the parameters including minimum, maximum, median, and mean, as shown in Figure 3. Therefore, the QTL hotspot on chromosome 4D was validated by marker1 “scaffold72468_211254” and marker 2 “scaffold72468_211948” to confer heat tolerance at the seedling stage in common wheat. However, for marker 3 and marker 4, no significant difference in phenotype was found. Therefore, only marker 1 “scaffold72468_211254” and marker 2 “scaffold72468_211948” have been validated as true markers for the associated QTL.

## 3. Discussion

This study identified QTL conferring HT at the seedling stage in common wheat based on root, shoot, whole plant length and their corresponding DIs. Variation in bi-parental populations is essential [32]. The phenotypic variations in the population were high for all the traits investigated in this study, which was fundamental for the successful identification of QTL conferring HT at the seedling stage. Mapping information of the ITMI population is available at GrainGenes database, and it has been successfully used for some QTL identification of different abiotic stresses in wheat [29,30,31]. Alsamadany investigated the diversity of heat tolerance performance of 499 genotypes with different origins at seedling stage under heat stress [33]. In his study, Opata 85 was identified as heat-tolerant at seedling stage, while synthetic W7984 was identified as heat-susceptible at seedling stage. Because of the high variation between the two parents of the ITMI population, we consider it a good material for identification and validation of QTL associated with heat stress.

Early stage phenotypic screening has been conducted on traits including plant fresh weight, dry weight, carbon isotope discrimination, canopy temperature, and spectral reflectance indices, etc., for the selection of tolerant genotypes [34]. We have previously investigated the relationship between heat tolerance at the seedling stage and reproductive stage and found significant correlations in response to heat stress between the two stages [18]. As the plant materials in this study were young and small, the variations in some traits such as plant fresh weight and dry weight were relatively low in general. In our previous study, it was found that there were variations in the seedling lengths, especially root lengths, among different wheat genotypes under heat stress condition [18], so we chose seedling length and corresponding DI for phenotypic screening in this study. DI is the ratio of the decrease in length under heat stress treatment to the length in the non-stress condition [33], it was extensively used in HT research and breeding. In this study, the QTL hotspot was identified by both the direct length under heat stress, and DIs derived from length related traits. Thus, seedling length, especially root length under heat stress, could be used as a selection criterion apart from DIs to evaluate HT performance at the seedling stage.

Six QTL conferring seedling heat tolerance with high LOD and R^2^ were identified on chromosome 4D with the same flanking SSR markers (Xmwg634 and Xbarc225). We searched the genes within this QTL hotspot, and in total, 572 genes with their function annotations (IWGSC Annotation v 2.1) were found (Appendix A); three genes, namely, “TraesCS4D01G009600LC.1”, “TraesCS4D01G018700LC.1”, and “TraesCS4D01G046000LC.1”, were described related to heat stress and are worth further focusing on. “TraesCS4D01G009600LC.1” is related to DNAJ heat shock N-terminal domain-containing protein, “TraesCS4D01G018700LC.1” is related to heat stress transcription factor A-9, and “TraesCS4D01G046000LC.1” is related to Class I heat shock protein [35]. The genomic region of 4D was also reported as a rich hub for genes controlling yield and yield-related agronomic traits in previous studies [36,37,38,39]. Some QTL were already identified on chromosome 4D as related to spike dry weight [34], grain yield [40], as well as plant height [38]. Cabral et al. [38] reported a significant QTL for the grain shape traits located on chromosomes 4D, accounting for up to 53.3% of the total phenotypic variation; in addition, similar to the situation in this study, several QTL associated with various traits were also found to locate at the same locus: For example, the most significant QTL for plant height, 1000 grain weight, and test weight were also detected on chromosome 4D at the same locus, suggesting that the hot spots on chromosome 4D may harbor some key genes related to yield in common wheat. It is notable that some major QTL clusters (≥15 individual QTL) were previously identified in MQTL regions on 4D, appearing to align with markers for dwarf gene Rht-D1 [41]. It suggests that the chromosome 4D region may play key roles in determining agronomic traits, which could affect all developmental stages including seedling and adult stages. Maulana et al. [6] previously reported that chromosomes 4A, 2B, 3B, 2D, and 7D harbor QTLs for heat tolerance of wheat at both seedling stage and adult stages.

The QTL hotspots, as genomic regions rich in QTL, are important since they may harbor key genes for the quantitative traits [42]. The introgression of such a QTL-hotspot region was reported to enhance drought tolerance and grain yield in chickpea cultivars [43].

A group of genes within an organism that were inherited together from a single parent is called a “haplotype”, and haplotype-based breeding has been regarded recently as a promising breeding approach [44]. In this study, the six major QTL in the QTL hotspot associated with different HT traits were all contributed by the tolerant parent Opata. QTL hotspot with a series of individual QTL clustered together may be the genetic region rich of favorable haplotypes. For example, a QTL hotspot on chromosome 2 in sweet cherry was used for positive selection of favorable haplotypes [45].

All the major QTL in this study and previously reported QTL on the same chromosomes identified under heat stress (HS) or heat and drought combined stress (HS + DS) were compared (Appendix A) [20,24,40,46,47,48,49], no overlapping was found. The closest distance was about 11.6 Mbp between QRlhti.uwa.3D in this study and MQTL3D.1 identified by Liu et al. [49]. We further compared the QTL hotspot in this study with previously identified yield-related genomic region on chromosome 4D (Appendix A) [38,40,47,49,50,51,52,53,54,55,56], it was found that some particular regions were within the identified locus in this study. These particular genomic regions include MTAs (marker trait association) associated with grain number [57], QTL associated with spike number and thousand-grain weight, and a gene (Rht2) associated with plant height [49], suggesting that the identified QTL hotspot in this study may be related to yield-component traits of common wheat.

Normally, the validation of the identified QTL is carried out by validating the flanking markers of QTL in other cross populations [30]. In this study, we used a different approach for QTL validation, which is validating the flanking SSR marker by other kind of marker (i.e., SNP marker), if the two kinds of markers could agree with each other for both the genotypic and phenotypic data, then the QTL was considered to be validated. Therefore, in this study, six QTL on chromosome 4D were identified and validated by genotyping-phenotyping association analysis using SNP assays.

A series of QTL related to wheat yield under abiotic stress were identified or validated by the same ITMI Synthetic/Opata population. For example, Onyemaobi et al. [30] identified and validated a major chromosome region for high grain number per spike under meiotic stage water stress. Two major QTL were detected on chromosome 5A when plants were exposed to water stress during meiosis, and one QTL was detected on chromosome 2A under normal watering condition. In another study, a high-density linkage map was constructed for seedling morphology under drought stress in common wheat by using synthetic/Opata population [29]. The map consisted of 2639 genotyping-by-sequencing markers and covered 5047 cm with an average marker density of 2 markers/cm. Moreover, 16 identified QTL explained 4 to 59% of the phenotypic variance. The QTL on 7B appeared to be the most significant QTL, explaining 59% of the phenotypic variance. An interesting phenomenon is that most of the positive alleles identified in the previous studies were contributed by parent Synthetic W7984, whereas for the current heat tolerance study, the positive alleles were mostly contributed by parent Opata, suggesting that different parents may contribute to resistance against different abiotic stresses, which emphasized the importance of germplasm diversity in practical breeding.

Much is unclear about the genetic mechanism of heat tolerance in wheat so far; therefore, it is important to further study HT at different developmental stages. A comparative study of the QTL between early stage and adult stage may provide a better understanding of the genetic mechanism of HT in wheat. The major QTL, especially the QTL hotspot, should be further studied for fine mapping and functional research for breeding of heat-tolerance.

## 4. Materials and Methods

### 4.1. Plant Material

A total of 111 RILs from ITMI mapping population derived from a cross between 2 common wheat genotypes—Synthetic W7984 (*Titicum turgidum* cv. Altar 84/*Aegilops tauschii* Coss. Line WPI 219) and Opata M85 were used under both non-stress and heat stress conditions. This population with genotyping and mapping information available in GrainGenes has been used for QTL identification in many studies [30,32,58]. Synthetic W7984 is heat-susceptible at seedling stage and Opata M85 is heat-tolerant at seedling stage [18,33].

### 4.2. Plant Growth and Treatment

A growth system that enabled evaluation of a large number of genotypes was developed using a customized hydroponic system [33] (Appendix A). Wheat seeds germinated were positioned and spaced 1cm apart in the middle of a filter paper or cloth on a horizontal line. The filter-paper/cloth inside the container was checked twice each day to make sure it was totally wet all the time. The bottom (about 2 cm high) of filter-paper/cloth was soaked in the water and the upper parts were given water-sprayed frequently to ensure there was no drought stress. This system was space-saving and convenient for carrying out treatment, non-destructive observation, and measurement. A high temperature of 35 °C was used for stress treatment, and an optimum temperature of 25 °C [59,60,61] was used as the control temperature. The plants ready for measurement after seven days growth.

Ten seeds for each genotype were soaked overnight in distilled water on a petri dish at 25 °C. Six uniformly germinated seeds were selected for each genotype and placed in the growth holder system and pre-moistened with distilled water. Three replications were used for each environment of control and heat treatment. The 2 environments were set as follows: 1 in a controlled environment room (CER) with 14-h photoperiod (200 μmol m^−2^ s^−1^), air relative humidity of 70.0–75.0% and constant temperature of 25 ± 1 °C, as the control; the other in similar conditions but at a temperature of 35 ± 1 °C, as the heat treatment. Light intensity was measured by light meter and adjusted to make it the same in the two environments. Distilled water matching the temperature was added to each container as needed throughout the evaluation period, to avoid water-deficit stress. The positions of the folders within each box were randomly changed every day to minimize random errors generated by the environmental differences within the box.

### 4.3. Phenotypic Evaluation and Heritability Estimation

The RILs were evaluated for shoot/root/whole plant length under heat stress and non-stress conditions. Root length (measured from the base of the crown to the tip of the longest root in cm), shoot length, as well as the whole length of individual plant were recorded after growing the plants for 7 days [59,60] in the phenotyping system.

Damaging indices (DIs) of shoot, root, and whole length, which were calculated with the formula of DI = $\frac{25{}^{\circ}C\;L\;-35{}^{\circ}C\;L}{25{}^{\circ}C\;L}\;$(L means root/shoot/whole length of plant) [33], were used as indicators to compare the HT performance of different genotypes. The lower the DI, the more tolerant the plant. Phenotypic variation was analyzed by a frequency distribution. Genotypic variation for heat-tolerance associated traits and their heritability were analyzed by Package of Variability particular for genetic variability analysis in Rstudio (Version 1.4.1106).

### 4.4. QTL Analysis and Mapping

Genotypic data of Synthetic/Opata population were obtained from GrainGene website (https://wheat.pw.usda.gov/GG3/, accessed on 2 May 2021). After phenotypic data were measured using the method described above, QTL analysis was performed by genotype-phenotype association analysis using WinQTL Cartographer v2.5 software. The locations and effects of QTL were determined following the composite interval mapping method (CIM) analysis [62]. The significant threshold LOD scores for QTL detection were determined based on 1000 permutations at *p* ≤ 0.05 [63]. The LOD peak location ≥3 was used to declare a QTL for heat stress [64].

### 4.5. Validation of Identified QTL

The identified major QTL were validated with a protocol as shown in Figure 4, using co-located SNP markers detected by Infinium™ Wheat Barley 40K v1.0 BeadChip (https://www.illumina.com/, accessed on 10 May 2021). Specifically, the sequence of flanking marker was searched by its name in NCBI database or GrainGene, and its physical position was identified by blasting with the wheat reference genome sequencing Ref V1.0 [35] (https://urgi.versailles.inra.fr/blast_iwgsc/?dbgroup=wheat_iwgsc_refseq_v1_chromosomes&program=blastn, accessed on 10 May 2021). SNP markers were searched by three selection criteria: (1) the SNP position was within the QTL interval or within 2 Mbp from each of the flanking markers, as SNP may often affect genes up to 2 Mbps away [65]; (2) the SNP was polymorphic between the two parents and thus among the RILs; and (3) the RILs possessing different alleles of the SNP showed significant difference in phenotypes, and based on that, the RILs can be divided into two groups, i.e., positive allele group or negative allele group. The SNPs meeting all the three selection criteira indicated that they were linked to the target QTL, and the polymorphic alleles contributed to the significant difference in the phenoypes. The identification of such SNP markers could therefore validate their linked QTL. The statistics of phenotypes was conducted by a one-tailed t-test, and the box plot was made by Rstudio (Version 1.4.1106).

## 5. Conclusions

Thirteen QTL including nine major QTL and four minor QTL were identified as related to growth traits and damage indices of common wheat under heat stress. A noteworthy QTL hotspot comprising six major QTL with the highest phenotypic variation was identified. It could not be identified under normal conditions but only identified under heat stress. The QTL hotspot was further validated by genotyping-phenotyping association analysis using SNP assays. The QTL and markers identified and validated in this study are useful information for marker assisted breeding of HT in common wheat.

## Figures and Tables

**Figure 1 plants-11-00729-f001:**
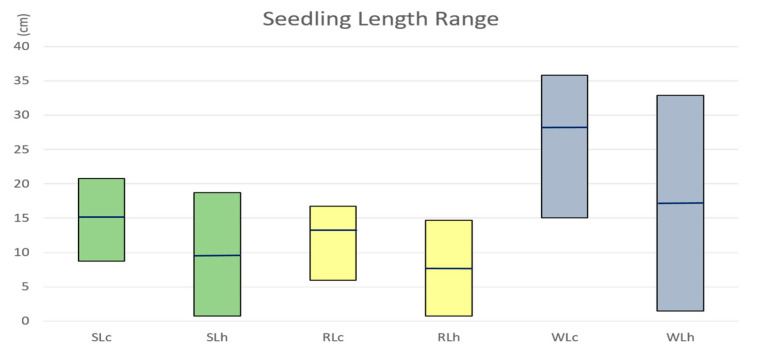
Comparison of seedling length under control and under heat stress. SL_c_, RL_c_, and WL_c_ indicate shoot length, root length, and whole seedling length under control, respectively; SL_h_, RL_h_, and WL_h_ indicate shoot length, root length, and whole seedling length under heat stress, respectively. Shoot, root, and whole length are shown in green, yellow, and blue colors, respectively. The top and bottom of the boxes show the minimum and maximum of length in the population, and the horizontal lines in the middle of the boxes show the average of length in the population.

**Figure 2 plants-11-00729-f002:**
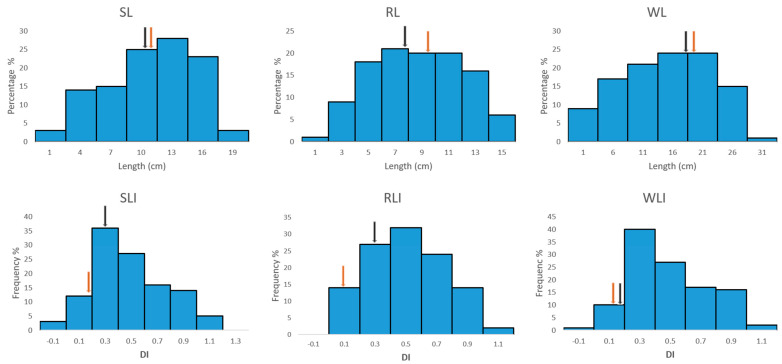
Frequency distribution of phenotypic variation for six traits among 111 RILs. SL, RL, and WL were shoot length, root length, and whole length values under 35 °C heat stress condition, respectively. SLI, RLI, and WLI were damage indices of shoot length, root length, and whole length, respectively. The values for Synthetic W7984 and Opata M85 parental lines are indicated by black and red arrows, respectively. Values shown are means.

**Figure 3 plants-11-00729-f003:**
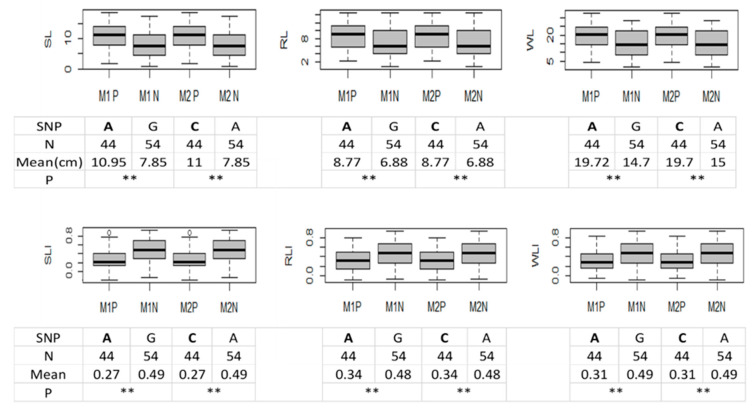
Match of SNPs with phenotype. SL, RL, WL indicate shoot length, root length, and whole plant length under 35 °C heat stress condition, respectively. SLI, RLI, and WLI indicate damage in-dices of shoot length, root length, and whole plant length, respectively. M1P and M2P indicate groups with positive alleles of marker1 and 2; M1N and M2N indicate groups with negative alleles of marker 1 and 2. SNP indicates the polymorphism allele. N indicates number of lines in the group. Totally, 339 individual plants (2 parents and 111 ITMI lines with 3 replications for each genotype) were grown and used for data collection. Only the lines with homozygotes were counted for phenotypic validation. *p* shows significant difference in phenotype between the groups with positive allele and negative allele. ** = significant at *p* ≤ 0.01. Polymorphic allele (positive allele followed by negative allele) and phenotype data analysis were below corresponding box plot for length traits and DIs. Two sample t-test assuming equal variance was used to detect statistical significance.

**Figure 4 plants-11-00729-f004:**
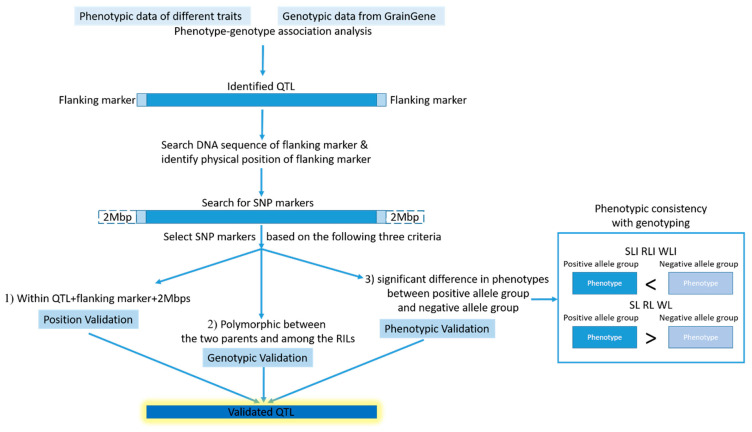
Protocol of validation for identified major QTL. The identified QTL was shown in dark blue and the flanking markers were shown in light blue. The searched area could be extended to 2 Mbp, showed by light blue dash lines. Phenotypic data of positive allele group were shown in dark blue and that of negative allele group was shown in light blue. The QTL validated by all the qualifications was shown by dark blue with yellow glow. SLI, RLI and WLI were shoot length, root length, and whole length index, respectively. SL, RL and WL were shoot length, root length, and whole lengths under 35 °C heat stress. SNP markers were searched by three selection criteria: (1) the SNP position was within the QTL interval or within 2 Mbp from each of the flanking markers, (2) the SNP was polymorphic between the two parents and thus among the RILs, and (3) the RILs possessing different alleles of the SNP showed significant difference in phenotypes.

**Table 1 plants-11-00729-t001:** Analysis of variance for heat tolerance associated traits and their heritability estimates in RILs.

Traits	MSg	MSe	$\delta_{g}^{2}$	$\delta_{e}^{2}$	$\delta_{p}^{2}$	h^2^
SL_c_	25.45 **	3.16	5.80	8.06	13.86	0.42
RL_c_	12.72 **	4.24	3.24	2.99	6.24	0.52
WL_c_	62.77 **	4.13	15.86	15.19	31.05	0.51
SL_h_	57.03 **	8.67	16.82	6.58	23.39	0.72
RL_h_	37.06 **	9.58	11.07	3.87	14.93	0.74
WL_h_	176.43 **	11.56	53.72	15.26	68.98	0.78

MSg: square of genotype; MSe: square of random error; $\delta_{g}^{2}$: estimated genetic variance; $\delta_{p}^{2}$: estimated phenotypic variance; $\delta_{e}^{2}$: estimated error variance; h^2^: heritability in broad sense. SL_c_, RL_c_, and WL_c_ indicate shoot length, root length, and whole length under control conditions, respectively; SL_h_, RL_h_, and WL_h_ indicate shoot length, root length, and whole length under heat stress, respectively; ** indicates significant difference at *p* < 0.01.

**Table 2 plants-11-00729-t002:** QTL identified under heat stress and control conditions in a Synthetic W7984/Opata M85 recombinant inbreeding population.

Trait	QTL Name	Position (cm)	Flanking Markers	LOD Score	Additive	Donor	R^2^
					Effect		
SLI	**QSli.4D**	18.2	Xmwg634/Xbarc225	12.45	0.16	Opata	0.32
RLI	Qrli.3D	22.5	Xbarc8/XksuA6	4.68	0.09	Opata	0.11
	**Qrli.4D**	18.2	Xmwg634/Xbarc225	11.61	0.15	Opata	0.33
	Qrli.6D	60.7	Xcdo534/Xgwm325	3.19	0.07	Opata	0.07
WLI	**Qwli.4D**	17.2	Xmwg634/Xbarc225	11.72	0.15	Opata	0.31
SL_h_	**QSlh.4D**	16	Xmwg634/Xbarc225	12.22	−2.56	Opata	0.33
	QSlh.5A	64	Xbarc151/Xbod183	4.1	1.38	Synthetic	0.09
	QSlh.6B	61.3	XksuG30/Xgwm219	3.31	−1.19	Opata	0.07
RL_h_	QRlh.2D	21.1	Xbod611/Xcdo1379	4.74	−1.24	Opata	0.11
	**QRlh.4D**	17.2	Xmwg634/Xbarc225	10.04	−1.81	Opata	0.26
	QRlh.5A	64	Xbarc151/Xbod183	3.3	1	Synthetic	0.08
WL_h_	**QWlh.4D**	17	Xmwg634/Xbarc225	12.32	−4.35	Opata	0.32
	QWlh.5A	64	Xbarc151/Xbod183	4.39	2.47	Synthetic	0.1
SL_c_	QSlc.2B	87.31	Xcdo678/Xmwg660	3.59	0.91	Synthetic	0.1
RL_c_	QRlc.2A	100.31	Xbarc353/Xgwm356	3.59	0.57	Synthetic	0.08
	QRlc.2B	51.31	Xgwm191/Xbcd1779	3.74	0.54	Synthetic	0.08
	QRlc.5A	87.41	Xbarc319/Xabg366	6.58	0.81	Synthetic	0.18
WL_c_	QWlc.2A	100.3	Xbarc353/Xgwm356	5.17	1.56	Synthetic	0.13
	QWlc.2B	47.11	Xbcd445/Xcnl6	3.52	1.24	Synthetic	0.09
	QWlc.5A	77.71	Xrz395/Xbarc230	5.33	1.87	Synthetic	0.2

SLI, RLI, and WLI are damage indices of shoot length, root length, and whole plant length, respectively. SL_h_, RL_h_, and WL_h_ indicate shoot length, root length, and whole plant lengths under 35 °C heat stress, respectively. SL_c_, RL_c_, and WL_c_ indicate shoot length, root length, and whole plant length under control, respectively. Each QTL is named with initial letter “Q” followed by trait abbreviation, and the corresponding chromosome. R^2^ means phenotypic expression. QTL with the highest R^2^ marked in bold.

## Data Availability

The data is contained within the article and Appendix A.

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
