# Peer review of "Identification and Validation of a Chromosome 4D Quantitative Trait Locus Hotspot Conferring Heat Tolerance in Common Wheat (Triticum aestivum L.)"

_plants, 2022, doi:10.3390/plants11060729_

Round 1
Reviewer 1 Report
In this manuscript 111 RILs from ITMI mapping population derived from a cross between Synthetic hexaploid wheat W7984 (Titicum turgidum cv. Altar 84/Aegilops tauschii Coss. line WPI 219) (a heat-susceptible variety at seedling stage) and Opata M85 (a heat-tolerant variety at seedling stage) were used for identifying and validating quantitative trait loci (QTL) responsible for heat tolerance (HT) in wheat based on the data of shoot length, root length, and whole seeding length under both non-stress and heat stress conditions. As a result, author s identified a chromosome 4D quantitative trait locus hotspot conferring heat tolerance to wheat. The story described in this manuscript is some interesting. The results achieved in this study are valuable for further fine mapping key genes, and exploring genetic mechanism of HT in wheat. Therefore, this paper can be considered for publication in this journal. I have two concerns as follows which need to be explained by authors before accepting.
1. High temperature is normally happened in wheat late grain-filling stage, which makes the flag leaf senescence and dramatically reduces the yield, not in seedling stage. Thereby, it should be more important to identify the QTLs conferring wheat with heat tolerance in adult stage other than seeding stage.
2. Authors only identified the QTLs associated with heat tolerances according to the phenotyping data of shoot length, root length, and whole seeding length, which are some simple. Why authors did not determined some physiological parameters related to heat tolerance?
3. To our knowledge, more than 180 RILs (at least 150 RILs) are normally used for the finding of QTLs in plants. But in this study authors only used 111 RILs for the QTLs on heat tolerance in wheat, and the population is some small.
Author Response
- High temperature is normally happened in wheat late grain-filling stage, which makes the flag leaf senescence and dramatically reduces the yield, not in seedling stage. Thereby, it should be more important to identify the QTLs conferring wheat with heat tolerance in adult stage other than seeding stage.
Answer: The reasons of this study on seedling stage are that:
1) Though generally reproductive stage is the most sensitive for final yield and the most studied stage [5], heat stress can affect the crop at different stages including seedling stage in some regions of the world [6]. Therefore, the study for seedling stage has practical meaning.
2) In our recent study, a significant positive relationship was established between heat tolerances at 7-day seedling stage and at adult stage [18].
3) Some reports indicated that some particular QTLs at seedling stage and reproductive stage were collocated on the chromosomes [13], moreover, some well-known heat-induced genes at reproductive stage were found significantly enhanced in acetylation levels in wheat seedlings [14]. Therefore, QTL identification and validation at seedling stage may help to understand the mechanism of heat tolerance at other stages.
(The references were marked the same numbers as in the manuscript.)
- Authors only identified the QTLs associated with heat tolerances according to the phenotyping data of shoot length, root length, and whole seeding length, which are some simple. Why authors did not determined some physiological parameters related to heat tolerance?
Answer: It was based on our previous studies that root length, shoot length, and whole seeding length were significantly affected by heat stress and the phenotypes showed significant variations. Various traits including some morphological traits and physiological parameters under stress were also studied in other previous studies. For example, Khalid et al. phenotyped the traits of Synthetic and Opata, including fresh weight, dry weight and length of root etc., the two parents showed differences in all the traits under abiotic stress (which has reference meaning for heat study), but the largest difference between the two parent lines was on root length [29]. This result was coherent with our studies [18], in which it was found that compared to other root traits such as root volume, root diameter and root shade area etc., root length showed the most variation among different genotypes under heat stress. Azam et al. [15] mapped QTL for physiological parameters such as chlorophyll fluorescence kinetics parameters at seedling stage of wheat under heat stress. Alsamadany [33] used seedling length as an indicator for heat tolerance. Therefore, we carried out this study on the basis of Alsamadany and focused on seedling length including shoot length and root length. We believe that the simpler the traits measured, the easier their application for wheat heat tolerant breeding. Although measurement of physiological parameters may help to understand better heat tolerance mechanism, it is beyond the scope of this research focusing on QTL mapping and validation.
(The references were marked the same number as in the manuscript.)
- To our knowledge, more than 180 RILs (at least 150 RILs) are normally used for the finding of QTLs in plants. But in this study authors only used 111 RILs for the QTLs on heat tolerance in wheat, and the population is some small.
Answer: The ITMI mapping population Sythentic/Opata has 115 lines with genotyping and mapping information available in GrainGenes. This population has been used for QTL identification in many studies [30, 32, 48]. We used 111 lines because four lines were lost during seed reproduction. Though the population size is a bit small, there were significant variations among the population lines and the phenotypic variation of the studied traits were normally distributed, so they were suitable for QTL analysis. Related information and reference of answers to this question have also been added in the manuscript in Materials and Methods (4.1 Plant material).
Reviewer 2 Report
This manuscript presents an interesting study on the presence of several QTLs associated with heat tolerance in the 4D chromosome of common wheat.
In general, the manuscript is well written and the data are adequately presented. Obviously, due to the numerous tracks observed in the manuscript, this has been previously send to other journal (Int. J. Mol. Sci.) and corrected according with the comments of other reviewers.
Some minor corrections should be carried out because some references are cited in the text in rare way. For example, the authors written in line 216: ‘In our previous study [18]’; however, this reference was authored by H. Alsamadany, who is not any of the authors of the current manuscript. Other references showed similar concordance problems. I suggest that the author should make a wide revision of the reference list and their mention in text for correcting these problems.
The authors written through the text: ‘wheat’, although they carried out all study with only one type of wheat, specifically ‘common wheat or bread wheat’ (Triticum aestivum L. ssp. aestivum). This should be clearly indicated in text.
Author Response
Some minor corrections should be carried out because some references are cited in the text in rare way. For example, the authors written in line 216: ‘In our previous study [18]’; however, this reference was authored by H. Alsamadany, who is not any of the authors of the current manuscript. Other references showed similar concordance problems. I suggest that the author should make a wide revision of the reference list and their mention in text for correcting these problems.
Answer: The above mentioned error has been corrected. The reference [18] has been replaced by our newly published journal paper with the same authors as this study, which is “Lu, L.; Liu, H.; Wu, Y.; Yan, G. Wheat genotypes tolerant to heat at seedling stage tend to be also tolerant at adult stage: the possibility of early selection for heat tolerance breeding, Crop J 2022, ISSN 2214-5141. (accepted)”.
All the references have been double-checked and other concordance problems have been corrected in the text and reference list.
The authors written through the text: ‘wheat’, although they carried out all study with only one type of wheat, specifically ‘common wheat or bread wheat’ (Triticum aestivum L. ssp. aestivum). This should be clearly indicated in text.
Answer: It has been clearly indicated through the text as suggested. The words “wheat” has been replaced by “common wheat” at all the necessary paces, including places in the Title, Abstract, Introduction, Results, Materials and Methods, and Conclusions.